# Mechanism Design and Experiment of a Bionic Turtle Dredging Robot

**Tao Wang** [1]**, Zhuo Wang** [2,*] **and Bo Zhang** [2]

1   School of Mechanical Engineering, Hebei University of Technology, Tianjin 300130, China; 1996075@hebut.edu.cn
2   College of Mechanical and Electrical Engineering, Harbin Engineering University, Harbin 150001, China; zhangbo_heu@hrbeu.edu.cn
*   Correspondence: wangzhuo_heu@hrbeu.edu.cn

**Abstract:** In order to clean underwater silt in artificially constructed rivers, lakes, and fish ponds, for which no suitable tool exists, a tool has been developed that imitates the structure and movement of the tortoise's legs, and designs a four-legged dredging robot that can adapt to the complex underwater environment. The article uses the transformation matrix to analyze the kinematics of the dredging robot, determines the movement sequence of the outriggers according to the principle of stability, and analyzes the movement characteristics of the three gait modes. Then, we combined the control function of the foot trajectory with the experimental prototype based on the bionic tortoise mechanism to carry out a walking experiment. During the experiment, the motion stability is good. Additionally, the changes in the position, the posture of the outriggers, and the body prove that the movement stability of the dredging robot using coordinated gait, mixed gait, and intermittent gait has increased sequentially.

**Keywords:** underwater silt; four-legged dredging robot; three gait modes; bionic tortoise mechanism; movement stability





## 1. Introduction

With the continuous development of industry, more and more garbage and waste are produced. Part of the polluted refuse is attached to the bottom of human-made rivers and lakes together with silt [1]. These silt and pollutants seriously affect the water quality of rivers and fish ponds, which also cause a large number of deaths of fish and other organisms in the water and destroy the ecological balance of the natural world [2]. In some rivers, the use of large dredgers for dredging operations can effectively clean up underwater silt and other pollutants [3,4]. However, in some fish ponds and small rivers, dredgers are inconvenient to start work and are not appropriate; therefore, there is a need for dredging equipment that can walk freely underwater [5,6].

Underwater dredging equipment can be divided into wheeled crawler, and foot equipment according to the walking mode underwater [7]. At present, wheeled and crawler dredging equipment is widely used because of their fast underwater movement speed and good stability [8]. However, these two dredging devices are difficult to adapt to complex environments [9,10].

Bruzzone and Quaglia summarize some of the motion characteristics of wheeled, crawler, and foot dredging equipment in the literature [11]. The foot robot can cross obstacles that are half of its own height. The ratio of the height of crawler and wheeled equipment that can cross obstacles to their own height is 0.25 to 0.5 and 0 to 0.25, respectively. The stair climbing capability and walking capability on uneven terrains of footed robots are also better than crawler and wheeled equipment.

Many underwater foot robots are quadruped robots [12]. In 2004, Boston Dynamics developed the simulation dog of quadruped robot, which can bear a rated load of 50 kg

on any terrain [13]. In 2010, Claudio, from the Italian Institute of Technology, developed the HyQ hydraulic quadruped robot, which weighs 70 kg and can crawl stably on rugged terrain [14]. In 2015, Boston Dynamics developed Spot, a light hydraulic quadruped robot weighing 72.6 kg, which is more flexible in movement. These four-legged robots have good movement ability on rugged terrain, but the load is small, and they are not suitable for underwater dredging work [15]. Therefore, it is very meaningful to develop a type of silt-removing robot that can move steadily on irregular terrain and carry a certain load at the same time.

In 1965, Hildebrand made the first major progress in quadruped gait. He described and compared the quadruped gaits of specific species based on physical characteristics and performance. In 1990, Zhang and Song formulated a wavy crab gait for a quadruped robot. The robot using this gait can move in any direction [16]. The current theoretical analysis of four-legged underwater robots is mainly its gait. The gait is defined by the lifting and placing time of the leg [17]. Later, the research on the gait of quadruped robots mainly focused on stability and anti-interference. For instance, in 2016, Chen et al., from the Wuhan University of Technology in China, proposed a gait planning method based on stability margin, which can realize a quadruped robot to travel stably on irregular terrain and select the optimal step distance [18]. In 2019, Boussema, from the Lausanne Institute of Technology and Bioengineering in Switzerland, used pulse sets to coordinate the adaptive lifting and landing of the quadruped robot's posture legs. This method can make the quadruped robot perform rapid interference recovery and excessive gait [19].

This article mainly analyzes the progress of the dredging robot from the perspective of walking stability, and according to a certain small-scale relationship, the model of foot-type underwater dredging robot is established so that it can walk on the road. The movement characteristics of different walking gaits are analyzed by observing the different postures and positions of the robot while walking.

## 2. Mechanism Analysis and Solution Method

### 2.1. The Overall Bionic Design of the Dredging Robot

In recent years, there have been many related studies on amphibious robots such as bionic crabs and bionic robotic shrimps [20,21], but their carrying capacity is far less than turtles that can carry turtle shells on their backs. Therefore, the tortoise, which can withstand high loads, has become a new bionic inspiration.

With reference to the structure of the tortoise's legs shown in Figure 1, the dredging robot also adopts a body with four legs, and each leg contains two outriggers. It can be seen in the tortoise skeleton diagram that the height of the turtle's plastron is lower than the joint between the hummus and the ulnar flexible bone of the front leg and the joint between the femur and tibia of the hind leg. This makes the tortoise's center of gravity lower and its movement more stable. Therefore, the two-section leg and the body of the dredging robot based on the bionic tortoise adopt this approximate triangle under the installation method.

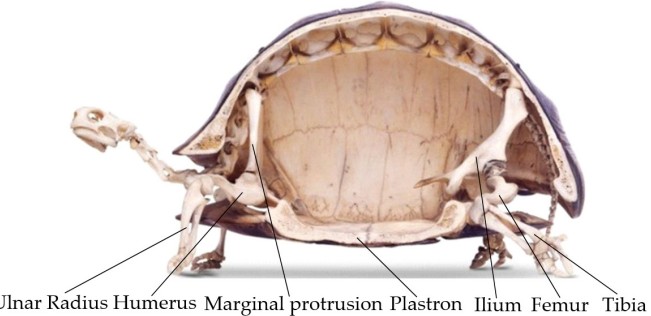

Ulnar Radius Humerus  Marginal protrusion  Plastron  Ilium  Femur  Tibia

**Figure 1.** Tortoise's bone structure.

The three-dimensional schematic diagram of the dredging robot is shown in Figure 2, which mainly includes the dredging device, outriggers, buoyancy device, main frame, and control system. LF, LH, RF, and RH represent the left front leg, the left hind leg, the right front leg, and the right hind leg, respectively. Each leg includes a supporting electric cylinder, a swing electric cylinder, and a supporting leg. The supporting electric cylinder can provide sufficient supporting force of the supporting leg to lift the body, and the swing electric cylinder can adjust the angle of the supporting leg relative to the body. There is friction between the foot and the support leg, which makes the angle between the foot and the support leg remain unchanged during the lifting process of the leg until it contacts the ground. After the foot landed, this angle is determined by the angle between the ground with the horizontal plane and the angle between the supporting legs with the horizontal plane.

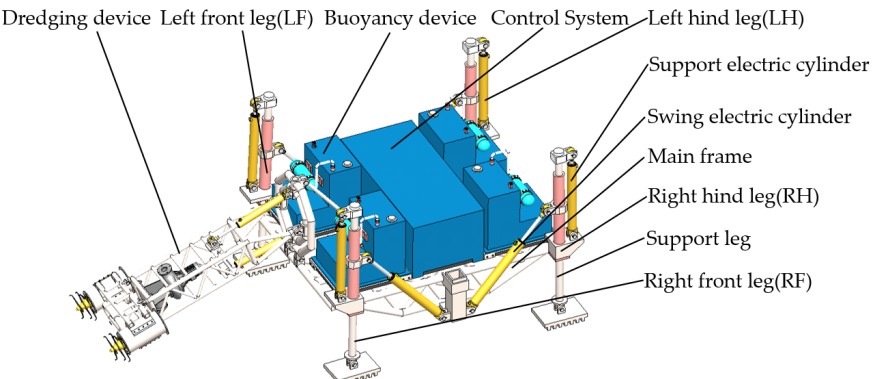

**Figure 2.** Three-dimensional schematic diagram of dredging robot.

The primary parts of the dredging robot are installed on the main frame. The leg is hinged with the main frame and is hinged on both sides of the middle of the main frame through the tail end of the swing electric cylinder. The sealed boxes of the four buoyancy devices are directly fixed on the upper layer of the main frame. The dredging device is hinged on the main frame, and the telescopic rod on the main frame can be tilted and swayed.

### 2.2. Outrigger Mechanism Modeling and Positive Kinematics Analysis Method

The four outrigger mechanisms of the dredging robot are geometrically symmetric about the fuselage, and therefore, only the fuselage and one outrigger are analyzed. The kinematics model of four outrigger mechanisms is established, as shown in Figure 3. We define the rotary joint between the supporting leg and the body as the hip joint, and the rotary joint between the supporting leg and foot as the ankle joint. $\beta_F$ is the angle between the ground and horizontal plane.

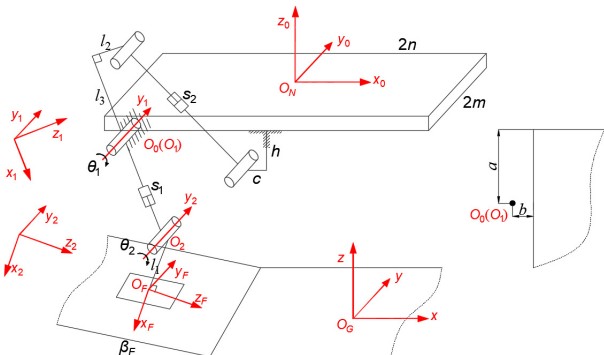

**Figure 3.** Kinematics model of dredging robot.

The carrier coordinate system $O_N$ of the dredging robot is established, and the direction of each coordinate axis of the dredging robot $O_N$ is exactly the same as that of the geodetic coordinate system $O_G$. The coordinate origin $O_N$ is the center of the upper layer of the dredging robot body, the $x_0$ direction is the same as the forward direction of the dredging robot, the $z_0$ direction is perpendicular to the plane of the body upward, and the $y_0$ direction is determined to point to the paper surface according to the right-hand rule. $O_0$ is the coordinate system where $O_N$ is translated from the upper center of the body to the hip joint, which is exactly the same as the direction of each axis of $O_N$. $O_0$ rotates $\theta_1$ around the $y_0$ axis to obtain the coordinate system $O_1$. The origins of the two coordinates coincide. At this time, the $x_1$ axis of $O_1$ points downward along the supporting leg. $O_1$ first rotates $\theta_2$ around the $y_1$ axis and then translates $s_1$ along the $x_1$ axis to obtain the ankle joint coordinate system $O_2$. The $x_2$ axis of $O_2$ is perpendicular to the ground and downward. $O_2$ is translated $l_1$ along the $x_2$ axis to obtain $O_F$, and the directions of each axis of $O_F$ and $O_2$ are exactly the same.

The length constant of the dredging robot is shown in Table 1. According to the three-dimensional model of the dredging robot, the highest position of the foot tip is 400 mm from the upper layer of the body, and the rotation range of $\theta_1$ is $(-0.75\pi, -0.25\pi)$. The angle of $\theta_2$ is determined by $\beta_F$ and $\theta_1$ together, $\theta_2 = 90° + \beta_F - \theta_1$.

**Table 1.** Kinematics model parameters of dredging robot.

| Parameter Symbol | Parameter Meaning | Value |
|---|---|---|
| $l_1$ | Vertical distance between ankle joint and foot plane | 160 mm |
| $l_2$ | The distance between the hinge point of the swing leg and the support leg and the hip joint in the vertical support leg | 220 mm |
| $l_3$ | The distance between the hinge point of the swing leg and the supporting leg and the hip joint along the supporting leg | 510 mm |
| $h$ | The vertical distance between the hinge point of the swing leg and the body and the plane of the body | 450 mm |
| $c$ | The vertical distance between the hinge point of the swing leg and the body and the center cross section of the body | 100 mm |
| $a$ | The distance between the hip joint and the short side of the body | 250 mm |
| $b$ | The distance between the hip joint and the long side of the body | 130 mm |
| $m$ | Half of the body length | 1700 mm |
| $n$ | Half of the body width | 1200 mm |

According to the D–H convention, the conversion matrix between $O_N$, $O_0$, $O_1$, $O_2$, and $O_F$ can be obtained, and then the transformation matrix between $O_N$ and $O_F$ can be obtained as shown in Equation (1).

$$
\begin{aligned}
{}^N T_F &= {}^N T_0 \cdot {}^0 T_1 \cdot {}^1 T_2 \cdot {}^2 T_F \\
&= \begin{bmatrix}
-\sin \beta_F & 0 & -\cos \beta_F & -l_1 \sin \beta_F + s_1 \cdot \cos(2\theta_1 - \beta_F) + a - m \\
0 & 1 & 0 & -n - b \\
-\cos(2\theta_1 - \beta_F) & 0 & -\sin(2\theta_1 - \beta_F) & -l_1 \cos(2\theta_1 - \beta_F) + s_1 \cdot \sin \beta_F \\
0 & 0 & 0 & 1
\end{bmatrix}
\end{aligned}
\tag{1}
$$

Using the above equation, we can obtain the coordinates ${}^N X_F$, ${}^N Y_F$, ${}^N Z_F$ representing the upper center of the body in the geodetic coordinate system, which are, respectively, equal to the three elements in the fourth column of the change matrix. Bring the above constants and variables into Equation (1) to obtain the foot's motion range, as shown in Figure 4.

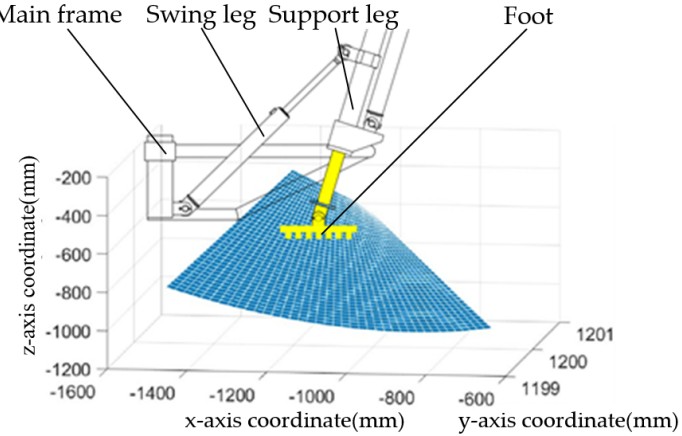

**Figure 4.** The trajectory and range of a single leg.

*2.3. Inverse Kinematics of Outrigger Mechanism and Method of Solving Body Pose*

In order to control the position and posture of the foot of the dredging robot in real time, it is necessary to obtain the function of the length of each leg and the expected posture of the foot. Since the $\beta_F$ value will be obtained after landing on foot, it is replaced with the $\beta_F$ on the same side of the foot.

According to Equation (1), it can be obtained as follow.

$$s_1 = \frac{-{}^N Z_F - l_1 \cdot \cos(2\theta_1 + \beta_F)}{\cos \beta_F}, \tag{2}$$

$$\theta_1 = \frac{\arccos \frac{{}^N X_F + l_1 \cdot \sin \beta_F + m - a}{s_1} + \beta_F}{2}, \tag{3}$$

According to the geometric relationship shown in Figure 5, the conversion formula between $\theta_1$ and variable $s_2$ can be obtained as shown in Equation (4).

$$\theta_1 = \arccos \frac{s_2{}^2 - l_2{}^2 - l_3{}^2 - (m - a - c)^2 - h^2}{2 \cdot \sqrt{(l_2{}^2 + l_3{}^2) \cdot [(m - a - c)^2 + h^2]}} - \varphi_1 + \varphi_2 = \arccos \frac{s_2{}^2 - 2.4703}{1.8992} - 0.8146° \tag{4}$$

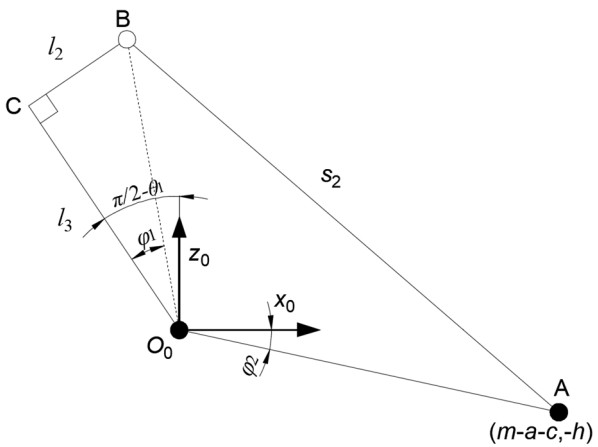

**Figure 5.** Analysis model of the single-leg mechanism.

Utilizing Equations (2)–(4) provides the analytical equations of variables $s_1$ and $s_2$, as shown in Equations (5) and (6), respectively.

According to Equations (1)–(3), the following can be obtained.

$$s_1 = \frac{-{}^N Z_F + \sqrt{{}^N Z_F{}^2 + 0.64 C\beta_F |{}^N X_F + 0.16 S\beta_F + 1.45|}}{2 C\beta_F} \tag{5}$$

$$s_2 = \sqrt{1.8992 \times \cos\left(\arccos\left(\frac{-{}^N X_F + 0.16 S\beta_F + 1.45}{s_1}\right)/2 + 0.8146°\right) + 2.4703} \tag{6}$$

In order to make the dredging robot have a proper step distance and the ability to cross obstacles, the displacement of the foot in the forward direction during a swing is increased by 1m, and the displacement in the vertical direction is increased by 0.3 m. According to the coordinate formulas of $^N X_F$ and $^N Z_F$, the coordinate range of the foot tip in the forward direction is $(-0.95, -1.95)$, and the coordinate range in the vertical direction is $(-0.9, -1.2)$. Substituting these values into Equations (5) and (6), the change diagrams of $s_1$ and $s_2$ corresponding to the coordinates of the foot in the advancing direction and the vertical direction can be obtained, as shown in Figures 6 and 7, respectively.

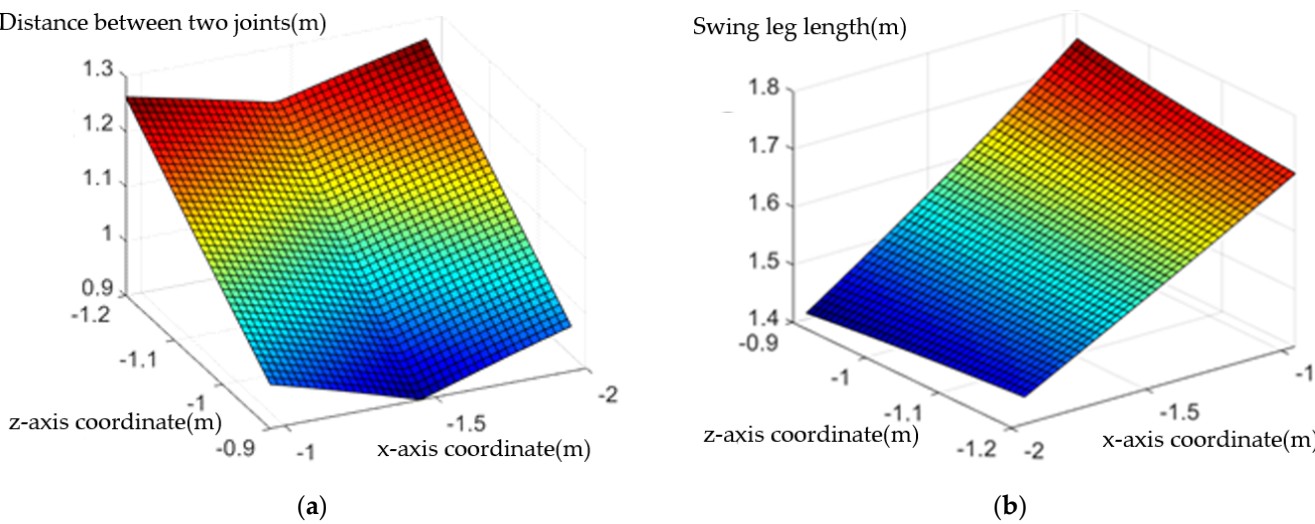

(**a**)                                             (**b**)

**Figure 6.** The length of the supporting leg and swinging leg in the desired coordinates: (**a**) foot coordinates corresponding to $s_1$ size and (**b**) foot coordinates corresponding to $s_2$ size.

Figure 6a shows that the range of the dredging robot $s_1$ is between 0.9 m and 1.3 m, that is, the change range of the length of the supporting telescopic mechanism is greater than 0.4 m. When the hip joint angle of the outrigger is not 90°, $s_1$ will increase to compensate for the z-axis coordinate change caused by the tilt of the outrigger, and the larger the $s_1$, the greater the absolute value of the z-axis coordinate.

Figure 6b shows that the range of the dredging robot $s_2$ is between 1.4 m and 1.8 m, that is, the swing telescopic mechanism and the supporting telescopic mechanism have the same length, and the change range must be greater than 0.4 m. The size of $s_2$ in the figure is mainly related to the x-axis coordinate and the closer the foot is to the center of the body on the x-axis, the longer the length of the swing mechanism.

According to the walking rules of the tortoise, at any time the dredging robot has at least a pair of angular side legs as supporting legs. When the LF and RH support the body, under the foot coordinate system of the LF, the coordinates of the hip joint of the LF and RH on the z axis are defined as $z_f$ and $z_h$, respectively. When the two feet are parallel to the horizontal plane, according to the geometric relationship in Figure 3, $z_f$ and $z_h$ can be obtained by Equation (7).

$$\begin{cases} z_f = l_{f1} + s_{f1} \cdot \cos \theta_{f2} \\ z_h = l_{h1} + s_{h1} \cdot \cos \theta_{h2} \end{cases}, \tag{7}$$

where $s_{f1}$ and $s_{h1}$ are the lengths of the hip joint of the LF and RH of the dredging robot from the bare joint, respectively; $l_{f1}$ and $l_{h1}$ are the vertical distance between the ankle joint of the LF and the RH and the supporting plane of the foot; $\theta_{f1}$ and $\theta_{h1}$ are the swing angles of the hip joints of the LF and the RH, respectively; and $\theta_{f2}$ and $\theta_{h2}$ are the swing angles of the ankle joints of the LF and the RH, respectively. When $l_{f1}$ and $l_{h1}$ are equal, the value of the pitch angle $\beta$ of the fuselage is calculated as shown in Equation (8).

$$\beta = \arctan \frac{s_{f1}\cdot \cos\theta_{f2} - s_{h1}\cdot \cos\theta_{h2}}{2m - 2a}, \tag{8}$$

According to Figure 3, when the RH of the dredging robot is in a supporting state, the coordinate equation of the gravity center of the machine can be calculated according to the coordinates of the hip joint relative to foot and the pitch angle of the body, as shown in Equation (9).

$$\begin{cases} {}^{F}X_N = -s_{f1}\cdot \sin\theta_{h2} + (m-a)\cdot \cos\beta \\ {}^{F}Y_N = n + b \\ {}^{F}Z_N = l_{f1} + s_{f1}\cdot \cos\theta_{h2} - (m-a)\cdot \sin\beta \end{cases}, \tag{9}$$

## 3. Gait Analysis and Trajectory Planning of the Bionic Turtle Robot

### 3.1. Turtle Crawling Mechanism

Figure 7a shows the coordinated gait of a tortoise. The four legs of the tortoise and their corresponding symbols are shown in Figure 7b. When the tortoise crawls, the center of gravity is in the stable area composed of limbs on the ground, and the step sequence relationship between the limbs of the triangular gait can be defined as 1—1′, 2—2′, 3—3′, 4—4′, as shown in Figure 7c [22].

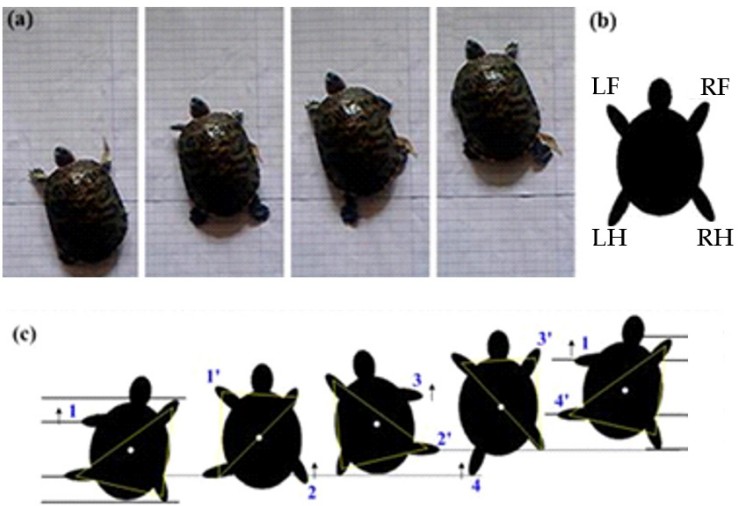

**Figure 7.** Turtle's crawling posture. (**a**) the coordinated gait of a tortoise, (**b**) The four legs of the tortoise and their corresponding symbols, (**c**) the step sequence relationship.

The crawling movement of the tortoise can be divided into eight stages. The relative positions of the limbs and the center of gravity in these stages are shown in Figure 8. In stage Figure 8a, LH, which has just landed, RF, and RH have established a triangular support area together, and the center of gravity falls within the triangle. At this time, LF is also on the ground; hence, there are two triangular support areas, and they partially overlap. When the LF moves forward in stage Figure 8b, it is steadily supported by a single three-legged support area. In stage Figure 8c, before the LF matches the ground, the RH lifts up, and the tortoise is temporarily supported by the opposite hind and front feet. In stage Figure 8d, LF reaches the ground and establishes a stable triangular support area again. The support patterns of stage Figure 8e, stage Figure 8f, stage Figure 8g, and stage Figure 8h are mirror images of the first four stages [23].

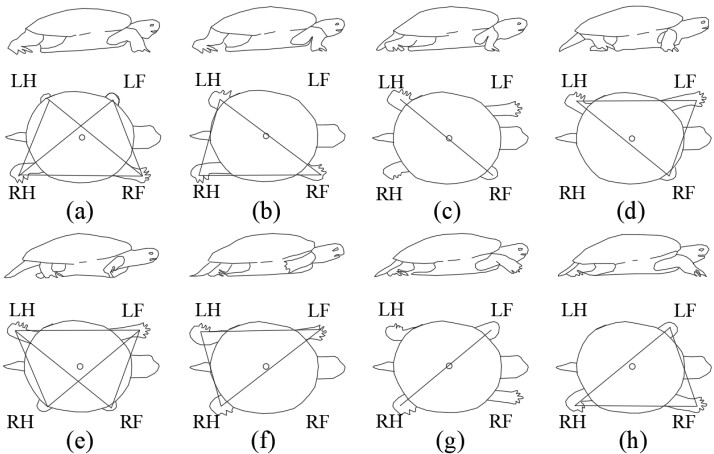

**Figure 8.** The eight crawling stages of a tortoise. (**a**) Before the LF moves, (**b**) the LF vacated, (**c**) before the RH moves, (**d**) the RH vacated, (**e**) before the RF moves, (**f**) the RF vacated, (**g**) before the LH moves, and (**h**) the LH vacated.

### 3.2. Gait Analysis of the Bionic Turtle Robot

### 3.2.1. Gait Sequence Design

According to the analysis method of permutation and combination, the four legs of the dredging robot have six moving sequences in total. If you follow the law of the animal's diagonal legs moving continuously, then there are four moving sequences. The gait diagrams of the dredging robot are shown in Figure 9.

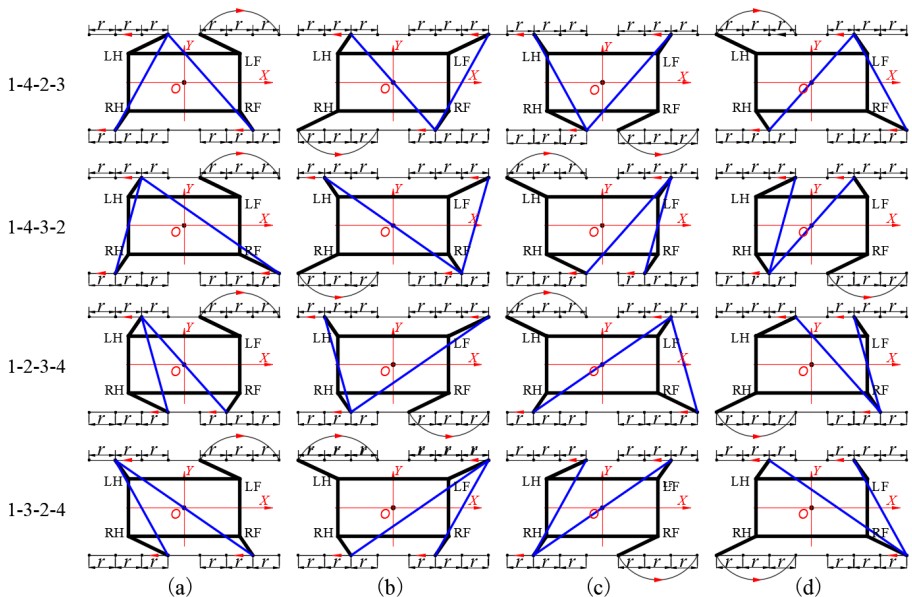

**Figure 9.** Gait diagrams in different sequences. (**a**) the moving of LH and RF, (**b**) the moving of LH, and RH, (**c**) the moving of LF, and RF, (**d**) the moving of LF and RH.

The triangle formed by the two blue straight lines in each gait diagram of Figure 9a–d and the straight line between the two feet on the same side represents the center of gravity stable area. When comparing the four moving modes, the 1–4–2–3 moving mode is the most stable because it has a sufficient stability margin. The *O* point of the center of gravity of the other three movement modes all have moments outside the triangle.

### 3.2.2. Three Kinds of Movement Gait of Dredging Robot

If the body is always supported by the four legs when the body moves forward, and the other three legs do not move when a single leg swings forward, then this gait mode

is an intermittent gait. It has high stability, but the flexibility of movement is relatively poor, and the speed is relatively low. It is suitable for walking on an irregular surface on the ground. The gait diagram of the dredging robot using intermittent gait motion is shown in Figure 10.

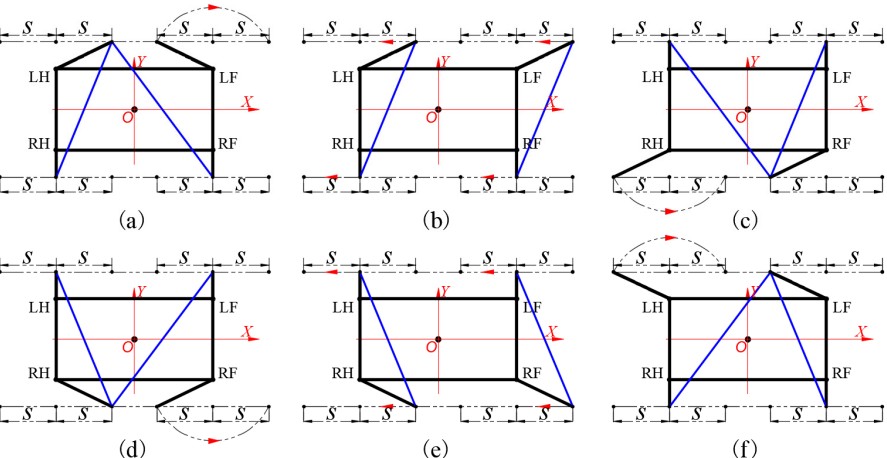

**Figure 10.** The relative position of the body and the ends of the four legs in the intermittent gait. (**a**) the LF has moved to the limit position, (**b**) the LF drops the ground, (**c**) the moving of LF, (**d**) the moving of RF, (**e**) the moving of RH and RF, (**f**) the moving of LF and LH.

The coordinated gait means that one leg is raised immediately when the other leg is on the ground, and the remaining three legs support the body to swing forward during the process of lifting one leg. Figure 10a–f of 1–4–2–3 shows a coordinated gait. In the state of Figure 10a, the LF has moved to the limit position, the remaining three legs support the dredging robot and swing to make the robot body move forward $f(r)$, and the three-legged feet move in the opposite direction of the x-axis relative to the center of gravity of the machine $r$, after which the LF drops the ground, as shown in Figure 10b. In this process, the stability margin gradually decreases to a critical state. At this time, the RH moves to the extreme position and moves forward, and then the LF and RF move forward at one time, as shown in Figure 10c,d.

The states shown in Figure 10b,d of the dredging robot in Figure 9 are critically stable states. After moving the position of the foot relative to the body when the robot is supported on four legs, the adjusted coordinated gait diagram is shown in Figure 11a–f.

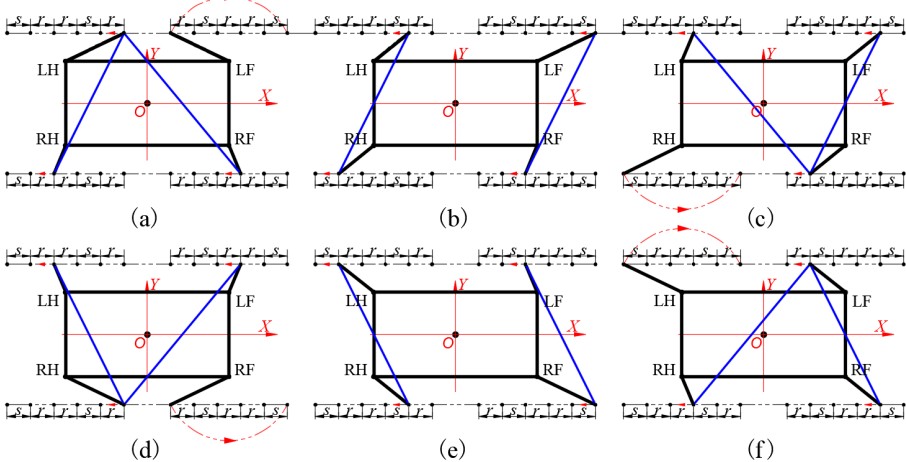

**Figure 11.** The relative position of the body and the ends of the four legs in the intermittent gait. (**a**) the LF has moved to the limit position, (**b**) the LF drops the ground, (**c**) the moving of RH, (**d**) the RH drops the ground, (**e**) the moving of LH, (**f**) the LH drops the ground.

Stages shown in Figure 11b,e are the adjustment actions of the mixed gait of the dredging robot relative to the increase in the coordinated gait. Figure 11b shows that after the LF of the dredging robot has landed and the RH is raised, all four legs support the body to swing forward, and the four-legged feet move a distance $S$ back relative to the center of the body. Figure 11e shows that after the RF has landed, and before the LH is raised, all four legs support the forward swing distance s of the body.

### 3.3. Motion Track Design

The maximum moving distance and maximum height of the foot in the forward direction during one swing are defined as E and H, respectively. When $t = 0$ and $t = T_0$, the forward direction of the foot should satisfy the position of 0 and $E$, respectively, and $T_0$ is the time of the single-leg swing. At the same time, in order to make the speed and acceleration of the piston rod of the electric cylinder more stable, the initial speed and the end speed of the foot in the forward direction are set to 0.

Using polynomial equations to plan the foot trajectory can control the speed of the foot at a specific point. Since the speed of the foot changes during the swing, the displacement function of the foot in the forward and vertical directions with respect to time is a cubic equation or a higher-order equation. In order to make the displacement function simpler, the displacement function with respect to time in the forward direction of the foot is defined as a cubic equation. Substituting the above foot position and time conditions into the equation, it can find the forward direction displacement equation of the foot as shown in Equation (10).

$$x(t) = -\frac{2E}{T_0^3}t^3 + \frac{3E}{T_0^2}t^2, \tag{10}$$

The movement of the foot in the z-axis direction should satisfy the position of 0, $H$, 0 when $t = 0$, $t = T_0/2$, and $t = T_0$. Then, the displacement equation of the foot tip in the vertical direction is shown in Equation (11).

$$z(t) = \begin{cases} -\frac{16H}{T_0^3}t^3 + \frac{12H}{T_0^2}t^2 & (0 < t < T_0/2) \\ H + \frac{16H}{T_0^3}\left(t - \frac{T_0}{2}\right)^3 - \frac{12H}{T_0^2}\left(t - \frac{T_0}{2}\right)^2 & (T_0/2 < t < T_0) \end{cases}, \tag{11}$$

Using time as a parameter, the trajectory of the foot during the lifting process can be obtained, as shown in Figure 12.

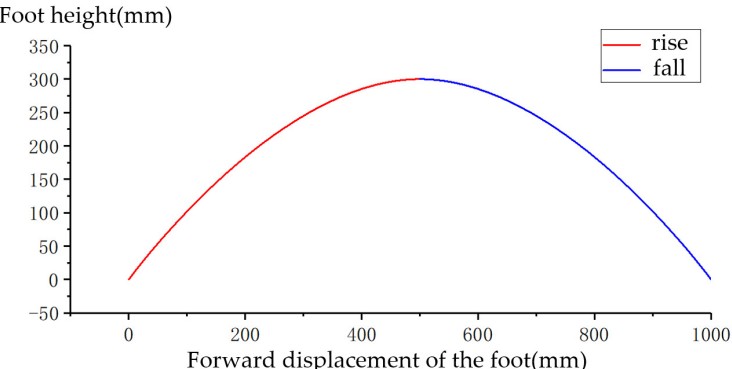

**Figure 12.** Foot trajectory.

The motion trajectory of the body needs to consider the motion trajectory of the remaining supporting legs in the motion state. Since the dredging robot does not move and rotate along the $y$ axis, it only needs the movement trajectory of one supporting leg at the front and hind. Taking the RF up at zero seconds as the time condition, the RH and the LF in 0—$T_0$ and $T_0$—$2T_0$ are all in a supporting state. At this time, the foot trajectory equation of the RH and the LF is shown in Equation (12). After $2T_0$, the LF is lifted, and the

RF and the LH in $3T_0$—$4T_0$ are all in a supporting state. At this time, the foot trajectory equation of the RF and the LH is shown in Equation (13).

$$\begin{cases} x_{\mathrm{RH}}(t) = E + \frac{2E}{27T_0^3}t^3 - \frac{3E}{9T_0^2}t^2 \\ x_{\mathrm{LF}}(t) = E + \frac{2E}{27T_0^3}(t+T_0)^3 - \frac{3E}{9T_0^2}(t+T_0)^2 \end{cases} (0 < t < 2T_0), \tag{12}$$

$$\begin{cases} x_{\mathrm{RF}}(t) = E + \frac{2E}{27T_0^3}(t-T_0)^3 - \frac{3E}{9T_0^2}(t-T_0)^2 \\ x_{\mathrm{LH}}(t) = E + \frac{2E}{27T_0^3}(t-2T_0)^3 - \frac{3E}{9T_0^2}(t-2T_0)^2 \end{cases} (2T_0 < t < 4T_0), \tag{13}$$

According to Equation (3), the relationship between $\theta_1$ and the abscissa of the foot can be obtained, and the relationship between $\theta_2$ and $\theta_1$ can be used to obtain the relationship between $\theta_2$ and the coordinate of the foot, as shown in Equation (14).

$$\theta_2 = 90° - \arccos\frac{x(t) + l_1 \cdot \sin\beta_F + m - a}{2s_1} + \frac{3}{2}\beta_F, \tag{14}$$

Take the above equation into the body's pose and coordinate solution equation. Since the front and hind support legs selected in $0$-$2T_0$ and $2T_0$-$4T_0$ are different, two fixed support legs need to be determined before $2T_0$ and after $2T_0$. In this way, the center trajectories of the front and hind ends of the body can be merged according to the coordinate relationship of the two fixed legs. In $0$-$2T_0$, select the foot of the RH as the reference coordinate origin of the body center coordinates and select the foot of the RF in $2T_0$-$4T_0$.

Substituting Equation (12) and Equation (14) into Equation (9), the equation for changing the pitch angle of the body within $0$-$2T_0$ can be obtained, and then substituting the obtained equation into Equation (9) calculates the center of the RH to obtain Equation (15).

$$\begin{cases} ^F X_N = -s_{f1} \cdot \sin\theta_{h2} + \frac{m-a}{\sqrt{1 + \left(\frac{s_{f1} \cdot \cos\theta_{f2} - s_{h1} \cdot \cos\theta_{h2}}{2m-2a}\right)^2}} \\ ^F Y_N = n + b \\ ^F Z_N = l_{f1} + s_{f1} \cdot \cos\theta_{h2} - \frac{s_{f1} \cdot \cos\theta_{f2} - s_{h1} \cdot \cos\theta_{h2}}{2\sqrt{1 + \left(\frac{s_{f1} \cdot \cos\theta_{f2} - s_{h1} \cdot \cos\theta_{h2}}{2m-2a}\right)^2}} \end{cases}, \tag{15}$$

In the $2T_0$-$4T_0$ time period, take the LF as a reference. According to the above method, the trajectory of the body center coordinate relative to the foot of the RF is obtained, and the trajectory of the body center in $0$-$4T_0$ can be obtained by combining the coordinate of the RH relative to the LF before lifting.

In the process of foot trajectory planning, the change of the foot position of the dredging robot with time is obtained, which is put into the solution equation of the RH $s_1$ and $s_2$, and the time change of the RH leg $s_1$ and $s_2$; the curves are shown in Figure 13a,b.

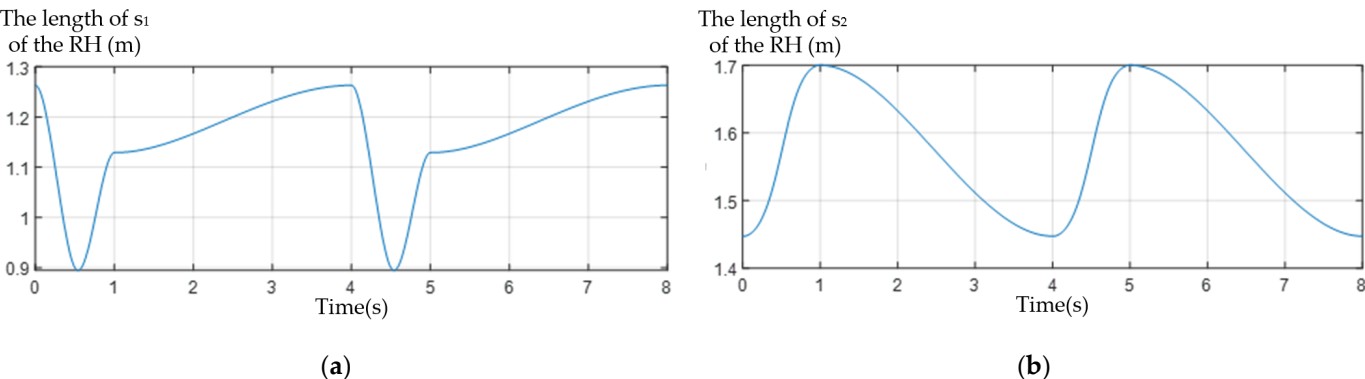

**Figure 13.** Changes in the length of the supporting leg and swinging leg in two cycles: (**a**) foot coordinates corresponding to $s_1$ size and (**b**) foot coordinates correspond to $s_2$ size.

## 4. Gait Experiment of Dredging Robot

Figure 14 shows the three-dimensional structure of the experimental prototype of the quadruped dredging robot. Compared with the actual underwater four-legged dredging robot, the experimental prototype retains the walking mechanism and the dredging frame installed with the dredging mechanism. In addition, the body of the original dredging robot is simplified, and a box made of aluminum alloy plates is used as the main installation frame of the prototype. The hardware control system of the outrigger and the hardware control system of the body are combined and all placed inside the box.

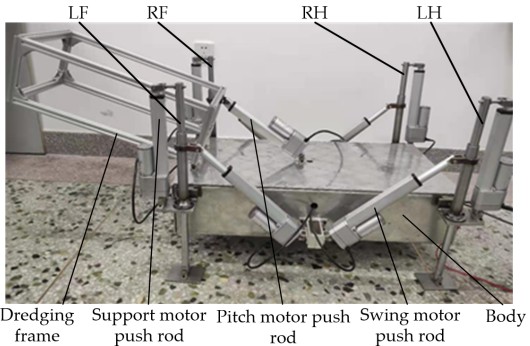

**Figure 14.** Experimental prototype.

The body size of the experimental prototype of the dredging robot is designed to be 800 mm in length, 400 mm in width, and 160 mm in height. Nine motor push rods are installed on it to replace the electric cylinder of the dredging robot in engineering, as shown in Figure 14. There are four support motor push rods and four swing motor push rods of LF, RF, LH, and RH, respectively, which serve as the support telescopic mechanism and the swing telescopic mechanism, and one pitch motor push rod is responsible for driving the rotation of the dredging frame. The power and signal lines on the motor push rod are connected to the power and signal lines in the box through the round hole on the box body.

### 4.1. Coordinated Gait Experiment

The experimental prototype of the dredging robot first conducts a coordinated gait motion experiment. In the experiment process, the motion trajectory equation of the foot of the experimental prototype was adjusted according to the relevant size parameters of the experimental prototype, and the motion equations of the eight motor push rods responsible for moving were obtained. The experimental prototype of the dredging robot is made to coordinate the gait movement with each cycle of 8 s, and each gait self-period is 2 s. After the posture is adjusted, it starts to move forward from the RH. Sampling is performed once in each subperiod during the movement, and the sampling pictures are shown in Figure 15a–d. Since there is no obvious change between the background and the ground, the demonstration effect of using a fixed camera to capture video is not ideal. Therefore, the geometric triangle and trapezoid changes represented by the red line can clearly express the foot movement.

The process shown in Figure 15a,b indicates the process of swinging of the RH and the remaining three legs supporting the body to move forward. During this process, the RH support motor push rod first extends and then shortens, and the swing motor push rod always extends. The support motor push rods of the remaining three legs have been shortened, the swing motor push rods of the two front legs are extended, and the swing motor push rod of the LH is shortened.

The process shown in Figure 15 (b) to (c), (c) to (d), and (d) to (a) indicates the forward swing process of the RF, LH, and LF. The change in the length of the four leg supports motor push rod during exercise is the same as that in Figure 15a,b. The change in the length of the swing motor push rod of the hind leg is also the same.

In the coordinated gait, the single leg of the experimental prototype swings from the hind limit position to the front limit position and moves 270 mm relative to the body, with a stability margin of 20 mm. In this process, the body moves 90 mm relative to the ground under the action of the remaining three legs, that is, the foot moves 360 mm relative to the ground. There is no four-legged support in the coordinated gait, and hence, the foot and the body move forward 360 mm in a cycle. The moving speed of the experimental prototype in the coordinated gait is 2.7 m/min, and the moving speed is converted according to the ratio of 1/5. The actual moving distance of the dredging robot per minute is more than 10 m.

Figure 16 is the change curve of the pitch angle of the experimental prototype body measured according to the gyroscope and the host computer software. It can be seen that the angle change range is between −3.5°~3.5°. Additionally, this can ensure that there is a sufficient safety distance between the bottom of the body and the ground during the movement.

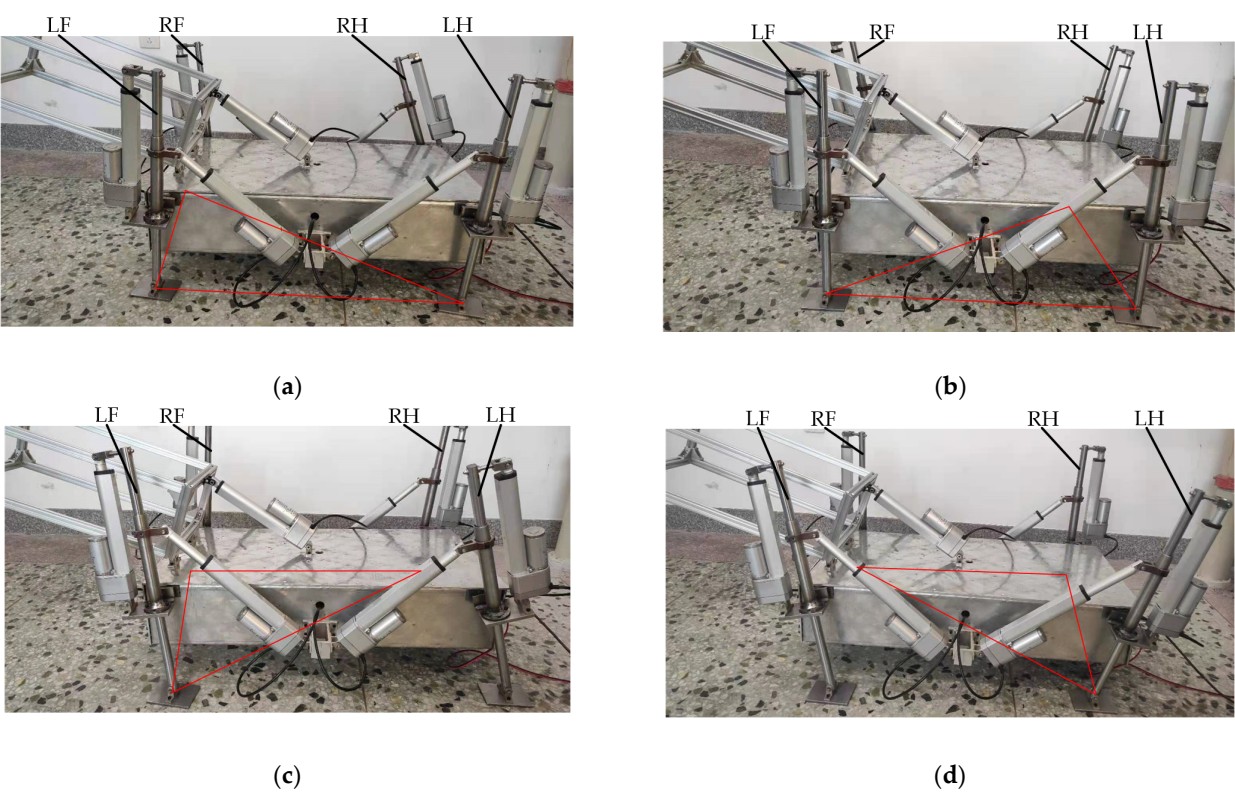

**Figure 15.** The motion state of the experimental prototype in a cycle in the coordinated gait: (**a**) the RH is about to lift, (**b**) the RF is about to lift, (**c**) the LH is about to lift, and (**d**) the LF is about to lift.

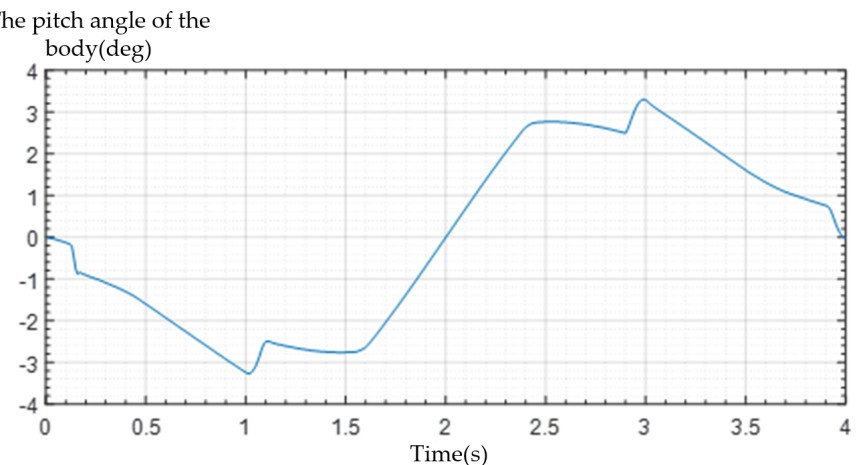

**Figure 16.** Pitch angle changes in coordinated gait.

### 4.2. Intermittent Gait Experiment

There are six gait subperiods when the experimental prototype travels in an intermittent gait. In the four gait subperiods, there are only single-leg swing motions with duration of 2 s, and two gait subperiods are four-legged support swing motions with duration of 1 s; therefore, the motion period of intermittent gait is 10 s. The change of the length of the telescopic mechanism during movement is relatively simple because each gait sub-period has three positions relative to the center of the body. The motor push rod is used to drive the experimental prototype of the dredging robot to walk, as shown in Figure 17a–f.

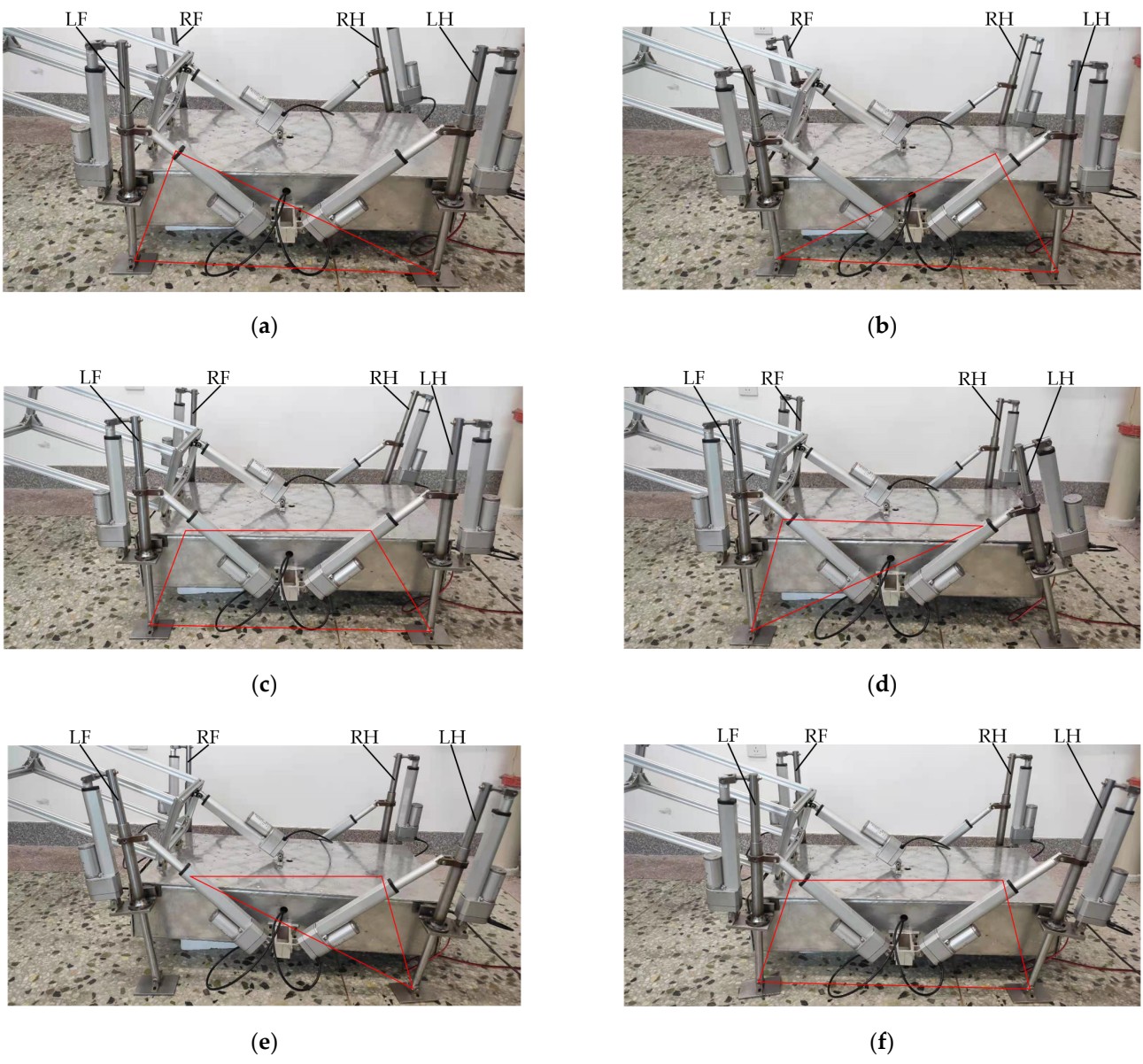

**Figure 17.** The motion state of the experimental prototype in a cycle in the intermittent gait: (**a**) the RH is about to lift, (**b**) the RF is about to lift, (**c**) adjust the center of gravity for the first time, (**d**) the LH is about to lift, (**e**) the LF is about to lift, and (**f**) adjust the center of gravity for the second time.

Figure 17a,b are the moments when the RH and RF are moved to the extreme position, and the rest of the outriggers maintain their current posture during the lifting process. The length change of the motor push rod of the swing leg is the same as the coordinated gait. The process shown in Figure 17c–e to a indicates the four-legged support adjustment stage of the experimental prototype to ensure that there is sufficient stability margin

during movement. During this process, the supporting motor push rods are all shortened. The swing motor push rod of the hind leg is shortened, and the swing motor push rod of the front leg is extended. Figure 17d,e show the state of the moment when the LH and LF are about to lift after the feet are moved to the extreme position.

According to Figure 17, compared with coordinated gait, the area of the stable support area formed by the end of the supporting leg of the intermittent gait is larger. The minimum stability margin during exercise is 50 mm, and the range of body inclination angle is $-2.5°$—$2.5°$.

The swing distance of a single leg relative to the body in the intermittent gait is 270 mm, during which the body is stationary relative to the ground. During the subsequent swing of the four-legged support, the body moved 135 mm relative to the ground. There are two four-legged support swings in one cycle of intermittent gait, and hence, the foot and body move forward 270 mm in one cycle. The moving speed of the experimental prototype in the intermittent gait is 1.62 m/min, and the actual forward distance of the dredging robot per minute is 8.1 m.

*4.3. Mixed Gait Experiment*

There are six gait subcycles when the experimental prototype travels in a mixed gait. The four gait subperiods are single-leg swing and the rest of the three-legged support swing action, for which the duration is 2 s, and the other two gait subperiods are the four-legged support swing action, for which the duration is 1 s, that is, the total duration of the mixed gait cycle is 10 s. The motor push rod is used to drive the experimental prototype of the dredging robot to walk, as shown in Figure 18a–f.

Figure 18a,b shows the moments when the RH and RF are moved to the extreme position, which is the same as the intermittent gait. The difference is that the remaining three legs support the body during the lifting process before swinging. The length change of all motor push rods during exercise is the same as the coordinated gait. The process shown in Figure 18a–f indicates the four-legged support adjustment stage of the experimental prototype. Figure 18d,e shows the moments when the LH and LF are about to lift after the feet are moved to the extreme position.

The area of the stable area of the supporting foot of the experimental prototype in the mixed gait is slightly smaller than that in the intermittent gait. The minimum stability margin during exercise is 40 mm, and the range of body inclination is $-3°~3°$, that is, stability is better than coordinated gait.

The swing distance of a single leg relative to the body in a mixed gait is 270 mm. During this process, the body moves 55 mm relative to the ground under the support of the remaining three legs, and the body moves 52.5 mm relative to the ground during the subsequent swinging process of the four-legged support. There are two four-legged supports and four three-legged supports in one cycle of mixed gait, so the foot and body move forward 325 mm in one cycle. The moving speed of the experimental prototype in the mixed gait is 1.95 m/min, and the actual forward distance of the dredging robot is 9.75 m/min.

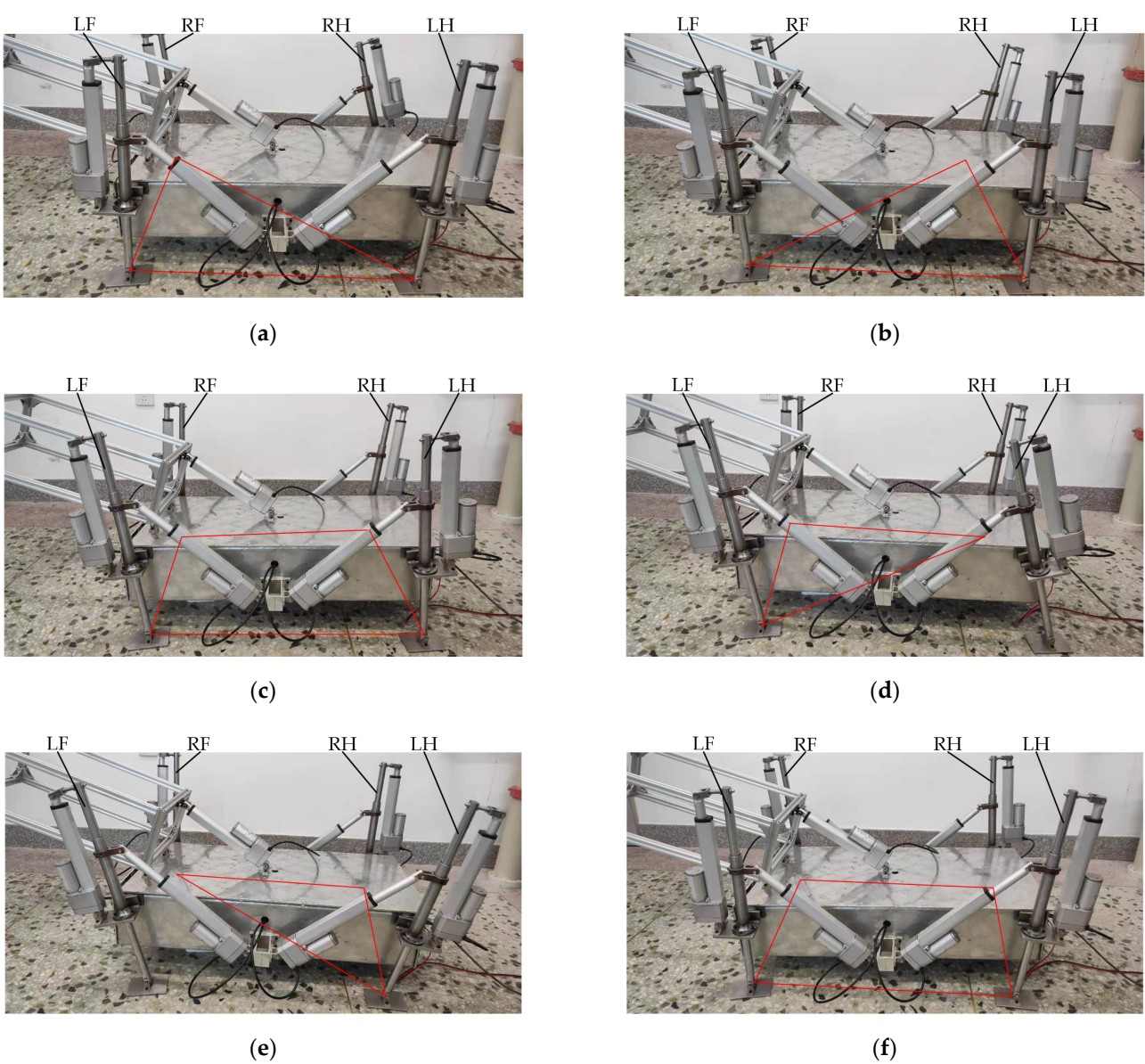

**Figure 18.** The motion state of the experimental prototype in a cycle in the mixed gait: (**a**) the RH is about to lift, (**b**) the RF is about to lift, (**c**) adjust the center of gravity for the first time, (**d**) the LH is about to lift, (**e**) the LF is about to lift, and (**f**) adjust the center of gravity for the second time.

## 5. Conclusions

This article refers to the tortoise's bone structure and movement. Based on this, the leg movement form and overall plan of the dredging robot are determined, and its movement gait is analyzed.

By establishing the kinematics model of the dredging robot, the transformation matrix between the body center and the foot, and the approximate fan-shaped motion space of the foot relative to the body center are solved. The moving range of the foot in the forward direction during a single swing is 0.7 m, and the moving range in the vertical direction is 0.8 m. additionally, through inverse kinematics, it is found that the variable length of the swing electric cylinder and the supporting electric cylinder of the outrigger is greater than 0.4 m.

This paper uses stability theory to determine that the stabilization of the dredging robot is the highest when it travels in the sequence of LF–RH–RF–LH. At the same time, the gait diagrams in different motion modes prove that the motion stability of the dredging robot in coordinated, mixed, and intermittent gait is improved.

This paper builds an experimental prototype of a 1/5 scale dredging robot. By using the foot trajectory planned by polynomial equations, the control program was written to make the experimental prototype move in coordinated gait, intermittent gait, and mixed gait. Experiments show that in coordinated gait, the minimum distance between the projection of the body center on the ground and the polygon formed by the end of the supporting foot is 5 mm, and the range of pitch angle of the body is $-3.5°$~$3.5°$; the minimum distance in intermittent gait is 100 mm. The range of pitch angle of the fuselage is $-2.5°$~$2.5°$, the minimum distance in mixed gait is 50 mm, and the range of pitch angle of the fuselage is $-3°$~$3°$. The experimental process and related data verify the feasibility of the dredging robot in three gaits and the accuracy of stability analysis.

The main purpose of the article is to explain the movement principle of a four-legged dredging robot. Since the experimental prototype is relatively simple, its movement process is not very smooth. Afterward, a more perfect experimental prototype or the actual product should be produced, and the accuracy of the kinematics and dynamics of the dredging robot should be verified.

**Author Contributions:** Conceptualization, T.W.; writing—original draft preparation, Z.W.; validation, B.Z. All authors have read and agreed to the published version of the manuscript.

**Funding:** This paper was funded by NSFC (Contract name: Research on ultimate bearing capacity and parametric design for the grouted clamps strengthening the partially damaged structure of jacket pipes). (Grant number: 51879063).

**Institutional Review Board Statement:** The study was conducted according to the guidelines of the NSFC and approved by Harbin Engineering University (protocol code 51879063 and date 1 January 2019).

**Informed Consent Statement:** Informed consent was obtained from all subjects involved in the study.

**Data Availability Statement:** We do not need to upload data. Readers can contact authors if they have questions. These authors can explain the data analysis.

**Conflicts of Interest:** The authors declare no conflict of interest.

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
