# Peer review of "Mechanism Design and Experiment of a Bionic Turtle Dredging Robot"

_machines, doi:10.3390/machines9050086_

Round 1

Reviewer 1 Report

This paper presented a bionic turtle robot for dredging. The robot modeling and gait analysis are detailed. It seems meaningful and useful. However, here are a few suggestions for further revision:

The author claims that the wheel and crawler dredging devices are difficult to adapt to complex environments. Without detailed data and comparison, this conclusion is hard to achieve. If the author can’t ensure the lift height and agility of the foot, the environment adaptability of the legged robot is also unreliable.

The contribution of this article is unclear. Please list the contributions and compare them with other robots.

Why the bionic design of turtles? Authors claim that “the smoothness of turtles when crawling is outstanding”, please provide the related references.

The size and parameters of robot including step size and height of leg lifting should be clear. “Assuming that the highest position of the foot tip is 400mm from the upper layer of the body, the rotation range of θ1 is (-0.75π, -0.25π).” It’s Inappropriate to assume the parameters.

The robot modeling and gait analysis are detailed, while the conclusion of the article is still week since the lack of comparing results between modeling and experimental. It’s better to provide more experimental data to make the conclusion more believable.

The locomotion demonstration of robot is not ideal. The video should be capture by a fixed camera and a locomotion movie or time-lapse pictures should be provided.

The grammar and style of writing have some errors throughout the manuscript. Some sentences are hard to understand for the reader and need to be corrected. Such as:

“In some large rivers and large rivers”

“Then the control function of the foot trajectory is combined with the experimental prototype based on the bionic tortoise mechanism, and then the walking experiment is carried out.”

“In this paper, by referring to the tortoise's bone structure and movement mode, the 420 leg movement form and overall plan of the quadruped dredging robot are determined, 421 and its movement gait is analyzed.”

Author Response

Responses to the reviewer 1—ID1146537

I am very grateful to your comments for the manuscript. According you’re your advice, we amended the relevant part in manuscript. Some of your questions were answered below.

To Reviewer #1:

1.The author claims that the wheel and crawler dredging devices are difficult to adapt to complex environments. Without detailed data and comparison, this conclusion is hard to achieve. If the author can’t ensure the lift height and agility of the foot, the environment adaptability of the legged robot is also unreliable.

Reply:A reference [11] is added to the introduction, which describes in detail some of the motion characteristics of wheeled, crawler and footed robots. In addition, some comparative explanations have been added to the text, which can prove that the foot-style exercise is more adaptable.

  1. The contribution of this article is unclear. Please list the contributions and compare them with other robots.

Reply:In the introduction, some classic quadruped robots are added, and the carrying capacity of these quadruped robots is too small, and a quadruped robot with a larger carrying capacity is needed for underwater dredging work.

  1. Why the bionic design of turtles? Authors claim that “the smoothness of turtles when crawling is outstanding”, please provide the related references.

Reply:The smoothness of the tortoise in the article has been modified to have a stronger carrying capacity compared to the lobster and crab.

  1. The size and parameters of robot including step size and height of leg lifting should be clear. “Assuming that the highest position of the foot tip is 400mm from the upper layer of the body, the rotation range of θ1 is (-0.75π, -0.25π).” It’s Inappropriate to assume the parameters.

Reply:These parameters are obtained from the three-dimensional model of the dredging robot, and the relevant description is added to the text, and the word hypothesis is deleted.

  1. The robot modeling and gait analysis are detailed, while the conclusion of the article is still week since the lack of comparing results between modeling and experimental. It’s better to provide more experimental data to make the conclusion more believable.

Reply:There is only one tilt sensor inside the experimental prototype that can feed back the tilt angle of the body. Because of the limited experimental equipment, it is difficult to measure the speed of the experimental prototype, the speed of the telescopic rod, and other data.

  1. The locomotion demonstration of robot is not ideal. The video should be capture by a fixed camera and a locomotion movie or time-lapse pictures should be provided.

Reply:The main purpose of the article is to explain the movement principle of a four-legged dredging robot. Because the experimental prototype is relatively simple, in fact, its movement process is not very smooth. I hope to take more accurate shooting and obtain more data after optimizing the experimental prototype, and I have added relevant instructions in the conclusion.

  1. The grammar and style of writing have some errors throughout the manuscript. Some sentences are hard to understand for the reader and need to be corrected.

Reply:The grammatical and stylistic errors in the text have been corrected.

************************************************

We would like to express our great appreciation to you and reviewers for comments on our paper. Looking forward to hearing from you.

Thank you and best regards.

Yours sincerely,

Zhuo Wang

Zhuo Wang

Eail: wangzhuo_heu@hrbeu.edu.cn

Apr.1.2021

Reviewer 2 Report

The paper presents the design and experiment results of a turtle-like robot for use in underwater tasks. The paper is very interesting, well structured, with adequate number of references and figures. However, there are some issues that should be addressed.

  • Section 2.2:
    • The figure 3 as well as the details presented in this section should be improved. What are the parameters and the variables of the mechanism? In fig 2 the variables are translational but the analysis is based on rotational angles \theta_1 and \theta_2.
    • In line 127 there is a reference about a transformation matrix that it is not shown. Is the kinematic analysis based on Denavit-Hartenberg convention?
    • How the orientation of the foot sole is changed? Is there any actuator?
  • Section 2.3:
    • The analysis in this paragraph is quite confusing.
    • In line 171 there is a small sentence which I don’t understand.
    • I cannot find z_h in figure 3.
    • Eq(9) refers to the same coordinate system shown in fig 3 or to another leg?
  • What are E and H in eq(10)-eq(13)?
  • The displacement function is a cubic one. Why?

Author Response

Responses to the reviewer 2—ID1146537

I am very grateful to your comments for the manuscript. According you’re your advice, we amended the relevant part in manuscript. Some of your questions were answered below.

To Reviewer #2:

  1. The figure 3 as well as the details presented in this section should be improved. What are the parameters and the variables of the mechanism? In fig 2 the variables are translational but the analysis is based on rotational angles \theta_1 and \theta_2.

Reply:The variables of the mechanism are θ1 and s1. θ2 can be calculated using βF and θ1. The θ2 of formula 1 in the article has been modified, and an explanation has been added in line 128 of the article.

  1. In line 127 there is a reference about a transformation matrix that it is not shown. Is the kinematic analysis based on Denavit-Hartenberg convention?

Reply:An explanation about the transformation matrix has been added to the text. The kinematic analysis is based on the D-H method, and an explanation is added in the text.

  1. How the orientation of the foot sole is changed? Is there any actuator?

Reply:The dredging robot has no actuators, and the rotation between the end of the foot and the supporting leg is passive. After the end of the foot has landed, θ2 is determined by the angle between the ground and the horizontal plane and the angle between the supporting leg and the horizontal plane. I added a description on line 94.

  1. In line 171 there is a small sentence which I don’t understand.

Reply:The wrong statement has been modified.

  1. I cannot find z_h in figure 3.

Reply:zh and zf are newly defined variables. I added a description and modified it on line 174.

  1. Eq(9) refers to the same coordinate system shown in fig 3 or to another leg?

Reply:It is the same coordinate system. I added a description in line 186 of the text.

  1. What are E and H in eq(10)-eq(13)?

Reply:The maximum moving distance and maximum height of the foot in the forward di-rection during one swing are defined as E and H, respectively. I added a description on line 254.

  1. The displacement function is a cubic one. Why?

Reply:Using polynomial equations to plan the foot trajectory can control the speed of the foot at a specific point. Because the speed of the foot changes during the swing, the displacement function of the foot in the forward direction and the vertical direction with respect to time is a cubic equation or a higher-order equation. In order to make the displacement function simpler, the displacement function with respect to time in the forward direction of the foot is defined as cubic equation. I added a description on line 260.

************************************************

We would like to express our great appreciation to you and reviewers for comments on our paper. Looking forward to hearing from you.

Thank you and best regards.

Yours sincerely,

Zhuo Wang

Zhuo Wang

Eail: wangzhuo_heu@hrbeu.edu.cn

Apr.1.2021

Reviewer 3 Report

The research of quadruped robot has a history of several decades. This paper aims at a kind of quadruped robot that imitates the tortoise's crawling, which belongs to relatively simple quadruped robot in terms of motion speed and motion mode. From the paper, we can see that we mainly use the kinematics and dynamics analysis, and verify the effectiveness of the proposed method through the experimental prototype. The work has a certain degree of difficulty and workload, but from the perspective of scientific and technological progress, the paper can not give the current status of other quadruped robots and quadruped robots in history.

Author Response

Responses to the reviewer 3—ID1146537

I am very grateful to your comments for the manuscript. According you’re your advice, we amended the relevant part in manuscript. Some of your questions were answered below.

To Reviewer #3:

  1. The research of quadruped robot has a history of several decades. This paper aims at a kind of quadruped robot that imitates the tortoise's crawling, which belongs to relatively simple quadruped robot in terms of motion speed and motion mode. From the paper, we can see that we mainly use the kinematics and dynamics analysis, and verify the effectiveness of the proposed method through the experimental prototype. The work has a certain degree of difficulty and workload, but from the perspective of scientific and technological progress, the paper can not give the current status of other quadruped robots and quadruped robots in history.

Reply:In the introduction, some classic quadruped robots are added, and the carrying capacity of these quadruped robots is too small, and a quadruped robot with a larger carrying capacity is needed for underwater dredging work.

************************************************

We would like to express our great appreciation to you and reviewers for comments on our paper. Looking forward to hearing from you.

Thank you and best regards.

Yours sincerely,

Zhuo Wang

Zhuo Wang

Eail: wangzhuo_heu@hrbeu.edu.cn

Apr.1.2021

Reviewer 4 Report

The paper provides the analysis of a bio-inspired dredging tool with the motion and stability based on a turtle.  I think the ideas of the paper are sound and the analysis is sound, but it is difficult to read.  There are a number of misused words and sections that don't make sense.  I suggest having the manuscript thoroughly reviewed for English usage before resubmittal.  I went thorough the initial parts of the paper in detail highlighting sections that need to be improved and adding comments along the way.  I did not identify all of the issues, but a number of them are noted. 

Once the English and grammar are fixed, it will be much easier to make a complete assessment of the technical content. 

Author Response

Responses to the reviewer 4—ID1146537

I am very grateful to your comments for the manuscript. According you’re your advice, we amended the relevant part in manuscript. Some of your questions were answered below.

To Reviewer #4:

  1. The paper provides the analysis of a bio-inspired dredging tool with the motion and stability based on a turtle. I think the ideas of the paper are sound and the analysis is sound, but it is difficult to read. There are a number of misused words and sections that don't make sense.  I suggest having the manuscript thoroughly reviewed for English usage before resubmittal.  I went thorough the initial parts of the paper in detail highlighting sections that need to be improved and adding comments along the way.  I did not identify all of the issues, but a number of them are noted.

Once the English and grammar are fixed, it will be much easier to make a complete assessment of the technical content.

Reply:

(1) The sentence errors pointed out in the article have been corrected.

(2) The description of gait in the introduction has been revised, which can reflect the technological progress of gait research.

(3) The symbol of the coordinate system has been modified.

(4) The 400mm data comes from the three-dimensional model of the dredging robot, and the relevant description has been added to the text.

(5) According to the analysis method of permutation and combination, the four legs of the dredging robot have six moving sequences in total. If you follow the law of the animal's diagonal legs moving continuously, then there are four moving sequences. These have been added to the text.

(6) It was pointed out that the unclear points in the text have been revised.

************************************************

We would like to express our great appreciation to you and reviewers for comments on our paper. Looking forward to hearing from you.

Thank you and best regards.

Yours sincerely,

Zhuo Wang

Zhuo Wang

Eail: wangzhuo_heu@hrbeu.edu.cn

Apr.1.2021

Round 2

Reviewer 1 Report

The author addressed the concerns partly, there are still some issues that need to be solved.

  1. The authors claim that their contribution is the introducing a quadruped robot with a larger carrying capacity, while I did not find any results and verifies about the carrying capacity in the main text.
  2. The robot modeling and gait analysis are detailed, while the conclusion of the article is still week since the lack of comparing results between modeling and experimental. It’s better to provide more experimental data to make the conclusion more believable.
  3. The locomotion demonstration of robot is not ideal. The video should be capture by a fixed camera and a locomotion movie or time-lapse pictures should be provided.

Author Response

Responses to the reviewer 1 again—ID1146537

I am very grateful to your comments for the manuscript. According you’re your advice, we amended the relevant part in manuscript. Some of your questions were answered below.

To Reviewer #1:

Opinion 1: The authors claim that their contribution is the introducing a quadruped robot with a larger carrying capacity, while I did not find any results and verifies about the carrying capacity in the main text.

Reply to: We didn't explain clearly the function of the quadruped robot in the original text. Now we will clarify its function. It is not a quadruped robot with greater carrying capacity, but can move steadily on irregular terrain and carry a certain load at the same time. , So, we modified the sentence of the article as follows: Therefore, it is very meaningful to develop a type of silt removing robot which can move steadily on irregular terrain and carry a certain load at the same time. See line number: 50-52, blue text.

Opinion 2: The robot modeling and gait analysis are detailed, while the conclusion of the article is still week since the lack of comparing results between modeling and experimental. It’s better to provide more experimental data to make the conclusion more believable.

Reply to: This article mainly analyzes the progress of the dredging robot from the perspective of walking stability, and establishes a foot-type underwater dredging robot model according to a certain proportion, so that it can walk on the road. The movement characteristics of different walking gaits are analyzed by observing the different postures and positions of the robot while walking. The kinematics analysis and solution method of bionic turtle dredging robot, gait analysis and trajectory planning of bionic turtle robot, and gait experimental analysis of robot were studied respectively. It can be concluded that: with the help of stability theory, it is determined that the stability of the dredging robot is the highest when it moves in the order of left foreleg right hind leg right foreleg left hind leg. At the same time, the gait diagrams of different motion modes prove that the motion stability of the dredging robot is improved in turn under coordinated gait, mixed gait and intermittent gait. The results show that in coordinated gait, the minimum distance between the projection of the body center on the ground and the polygon formed by the supporting foot is 5mm, and the range of the fuselage pitching angle is - 3.5 ° ~ 3.5 °; in intermittent gait, the minimum distance is 100 mm, and the range of the fuselage pitching angle is - 2.5 ° ~ 2.5 °; in mixed gait, the minimum distance is 50 mm, and the range of the fuselage pitching angle is - 3 ° ~ 3 °. The experimental process and related data verify the feasibility of the dredging robot in three kinds of gait and the accuracy of stability analysis.

Opinion 3: The locomotion demonstration of robot is not ideal. The video should be capture by a fixed camera and a locomotion movie or time-lapse pictures should be provided.

Reply to : Because there is no obvious change between the background and the ground, the demonstration effect of using fixed camera to capture video is not ideal. Therefore, the geometric triangle and trapezoid changes represented by the red line can clearly express the foot movement, and explain the robot foot movement process in the text. See line number: 341-343, blue text. Figures 15, 17 and 18.

************************************************

We would like to express our great appreciation to you and reviewers for comments on our paper. Looking forward to hearing from you.

Thank you and best regards.

Yours sincerely,

Zhuo Wang

Eail: wangzhuo_heu@hrbeu.edu.cn

Apr.19.2021

Reviewer 2 Report

The authors addressed all my comments. One last issue that should be addressed is the following:

Add a table with the D-H parameters in page 4 after table 1.

Author Response

Responses to the reviewer 2 again—ID1146537

I am very grateful to your comments for the manuscript. According you’re your advice, we amended the relevant part in manuscript. Some of your questions were answered below.

To Reviewer #2:

Opinion 1: Add a table with the D-H parameters in page 4 after table 1.

Reply to: Since the leg of the quadruped robot is not a series mechanism, but only a single joint motion, it is unnecessary to fill in the D-H parameter table. In this paper, the motion of this joint has been shown in line 130-138. Formula (1) and Figure 4 can clearly explain the trajectory and range of a single leg. See line number: 130-138, blue text.

************************************************

We would like to express our great appreciation to you and reviewers for comments on our paper. Looking forward to hearing from you.

Thank you and best regards.

Yours sincerely,

Zhuo Wang

Eail: wangzhuo_heu@hrbeu.edu.cn

Apr.19.2021

Reviewer 4 Report

With the second submission, some sections still need to be edited/reworded.  There are very polished sections and other sections that still need work in terms of English grammar and clarity. 

Attached are comments for additional consideration. 

Author Response

Responses to the reviewer 4 again—ID1146537

I am very grateful to your comments for the manuscript. According you’re your advice, we amended the relevant part in manuscript. Some of your questions were answered below.

To Reviewer  #4:

Opinion 1: Attached are comments for additional consideration.

Reply to: Reply to: According to the reviewers' comments, we have revised the article again. See the blue text. Please review it again.

************************************************

We would like to express our great appreciation to you and reviewers for comments on our paper. Looking forward to hearing from you.

Thank you and best regards.

Yours sincerely,

Zhuo Wang

Eail: wangzhuo_heu@hrbeu.edu.cn

Apr.19.2021
